# Effects of Dietary Supplementation of Acetate and L-Tryptophan Conjugated Bypass Amino Acid on Productivity of Pre- and Post-Partum Dairy Cows and Their Offspring

**DOI:** 10.3390/ani11061726

**Published:** 2021-06-09

**Authors:** Jang-Hoon Jo, Jae-Sung Lee, Jalil Ghassemi Nejad, Won-Seob Kim, Jun-Ok Moon, Hong-Gu Lee

**Affiliations:** 1Department of Animal Science and Technology, Sanghuh College of Life Sciences, Konkuk University, Seoul 05029, Korea; godandthegod@naver.com (J.-H.J.); jslee78@konkuk.ac.kr (J.-S.L.); jalilgh@konkuk.ac.kr (J.G.N.); kws9285@hanmail.net (W.-S.K.); 2Institute of Integrated Technology, CJ CheilJedang, Suwon 16495, Korea; junok.moon@cj.net

**Keywords:** blood hematology, bovine, dry cow, metabolism, tryptophan, newborn calves

## Abstract

**Simple Summary:**

This study examined the effect of acetate and L-tryptophan-conjugated bypass amino acid (ACT), supplemented (15 g/day) to Holstein cows during late pregnancy, on their productivity and the performance of offspring. We identified that the supplementation of ACT incorporated into diet was beneficial for improving the feed intake, blood hematology, and metabolites of the prepartum, and also had a positive effect on reducing saturated fatty acids in the colostrum of the cows postpartum and on the body weight of the newborn calves. The results of this study suggest that ACT supplementation improves the productivity of dairy cows.

**Abstract:**

In this study, we investigated the effect of dietary supplementation with acetate and L-tryptophan-conjugated bypass amino acid (ACT) during late pregnancy on the production performance of cows pre- and postpartum and their offspring. Eight multiparous Holstein cows (calving date ±15 d, 2nd parity; *n* = 4) were supplied with diets without ACT supplementation (Control) or with 15 g/day ACT supplementation (ACT). The results showed that ACT improved the feed intake (FI) in dry cows. No differences in blood hematological parameters were found between the two groups of prepartum cows. The serum glutamic-oxaloacetic transaminase activity increased and the triglyceride concentration decreased in the ACT-treated group compared to the control group. In the postpartum cows, milk compositions were not affected by ACT supplementation. Saturated fatty acid (SFA) content in the colostrum was significantly lower in the ACT-treated group than in the control group. Serum glucose (GLC) level was significantly higher in the ACT-treated group than in the control group. Monocyte and GLC levels were lower in calves of groups where their dams had received ACT. Overall, we found higher FI in the dry cows, lower colostrum SFA levels, and heavier calf birth weight (5.5 kg) when the dams were supplemented with ACT, suggesting a positive nutrient compensation by ACT supplementation to dry cows.

## 1. Introduction

The dry period in dairy cows is a pivotal phase for the development of resistance to diseases such as mastitis, for improving productivity postpartum, strengthening the body tissues for parturition, and promoting calves’ health postpartum [1], especially because of the rapid growth of the fetus and uterus during the last trimester of the pregnancy. The digestive tract is also pressurized and this results in reduced feed intake (FI) due to endocrine changes that are necessary for parturition and lactation, which may cause hormonal imbalances in the body [2]; a natural genetic change in the hormonal cycle that appears only during the dry period [3]. In terms of epigenetic control, dairy cows are inappropriately monitored during the dry period for milk production and reproductive performance, which appears to negatively affect their lactation postpartum. The lack of dairy cow monitoring during the dry period can also affect fetal and postnatal calf growth. In this respect, supplementation with necessary nutrients during the dry period is an important consideration for improving offspring growth and milking after calving.

L-tryptophan is an essential amino acid (EAA) in ruminants and a necessary component of protein synthesis, and also acts as a precursor of many neurotransmitters, including serotonin, melatonin, and niacin [4,5]. The lack of L-tryptophan adversely affected FI and growth performance in beef cattle [6,7] and thus needed to be validated in dairy cows. Furthermore, the need for L-tryptophan for protein synthesis by the fetus suddenly increases while at the same time FI decreases during the dry period [8]. Given this phenomenon, the feed of dry cows should be supplemented with L-tryptophan, and its effect on dry cows as well as the carry-over effects on milk production postpartum, and a possible compensatory effect on offspring performance and health status, need to be evaluated. L-tryptophan is a nutritionally important substance and has been reported to enhance the secretion and activity of pancreatic alpha-amylase to boost starch digestibility by enhancing the secretion of the gastric mucosa and stimulation of the glandular hormone cholecystokinin (CCK) in the small intestine [4].

When L-tryptophan is supplemented to ruminants, it is used by rumen microorganisms to increase microbial protein synthesis [9]. Thus, not only large amounts of un-utilizable ammonia are produced, but also the utilization rate of the feed protein is decreased. Therefore, it is necessary to feed bypass L-tryptophan to avoid microbial degradation in the rumen and enhance absorption in the small intestine. In this regard, to protect the feed protein from ruminal microorganisms, heat treatment [10] and coating methods with fatty acid/pH-sensitive polymer mixtures or coating with unsaturated fatty acids and mineral mixtures have been used. Moreover, with the increasing economic value of milk protein, dairy producers and industries have had an increasing interest in nutritional modifications, such as supplementation with rumen-protected AA, which increases the milk protein yield, and other components related to milk quality [11]. Tryptophan bound to acetyl (N-acetyl-L-tryptophan, ACT) is absorbed in the small intestine by a 95% bypass of the rumen [6]. Feeding dietary ACT has been shown to increase the growth performance of lambs [12] and the average daily gain (ADG) and N in cashmere goats [5].

Therefore, the objectives of this study were to investigate the effect of supplementing ACT to dry cows during late pregnancy on the FI and milk characteristics of lactating dairy cows postpartum, as well as evaluating its nutritional compensatory effects on the weight and health status of newborn calves, by examining their production performance and blood characteristics.

## 2. Materials and Methods

### 2.1. Experimental Design

This study was conducted at the Konkuk University experimental farm, Chungju City, Republic of Korea. All procedures were evaluated and approved by the Institutional Animal Care and Use Committee at Konkuk University. Eight Holstein cows with the same parity (second parity; *n* = 4 for each of the 2 groups) were dried off at about 60 d (day 245 to day 305 of gestation) before the expected calving date (calving date ± 15 d) and at a similar body weight (650 kg ± 18, *p* > 0.10). Since the expected parturition days did not match at exactly 60 days, the experimental period was 42 ± 8 days. The control group was fed a basal diet and the ACT-treated group was a fed basal diet supplemented with an additional 15 g of ACT (CJ CheilJedang, Suwon, Korea). The ACT was in the form of acetate and L-tryptophan, which had a bypass efficiency of 95% [7]. After feeding with ACT, the amount of ACT fed was gradually increased by 5 g, from 10 g up to 15 g from 0 weeks to 1 week before starting the experiment. The ACT was provided daily as a top-dressing of the corresponding experimental diet at 09:00 and 15:00.

### 2.2. Basal Diet and Colostrum Feeding

Basal diets for the dry and milking period were fed according to NRC nutrient requirements (2001; Table 1). The basal diet and amino acid (AA) compositions are summarized in Table 1. The chemical compositions of the feed, including crude protein, ether extract, amino acids, were analyzed according to the AOAC (Official Methods of Analysis, 15th Edition, 1991). Colostrum was collected from the cows within 3 h after calving, and 50 mL of the sample was frozen immediately after collection and stored at −20 °C for composition analysis. Each calf received colostrum [8] from its respective dam at 8% of their body weight (BW) within 3 h of birth by an esophageal feeder, and water was provided ad libitum. After three days, calves were fed pasteurized milk for eight weeks at 8% of their BW [8] and ad libitum feeding of starter diet until weaning at eight weeks.

### 2.3. Individual Animal Sampling and Analysis

#### 2.3.1. Feed Intake Measurement

The feed and ACT were given to each of the four individual dry cows twice a day, at 09:00 and 15:00 for 60 d as a top-dressing on the TMR. The remaining four cows served as controls. The FI of the cows in each group was determined using a weighing scale (GL-6000S Series, G-Tech International Co., LTD, Uijeongbu, Seoul, Korea) and by calculating the cumulative weights of all feed delivered minus the refusals. The dry matter intake (DMI) for each group was recorded once a day (08:30) by weighing the orts, calculated using the dry matter (DM) of all diets fed to each group before providing the feed.

#### 2.3.2. Birth Weight and Weight Changes of Newborn Calves

All calves (*n* = 8; 4 animals per treatment) were weighed at birth using a weighing scale (GL-6000S Series, G-Tech International Co., LTD). Additionally, the BW of the calves was obtained every two weeks until weaning at eight weeks. After parturition, the colostrum from each cow (from both the morning and evening milking) was collected for three consecutive days and provided individually to the corresponding calves. The calves were housed in individual wire hutches (1.5 m × 2.5 m) bedded with sand and managed in the same manner thereafter. Three male and one female calves were born in the control and treatment groups, respectively.

#### 2.3.3. Milk Yield

After parturition, milk yield was recorded daily (03:00 and 15:00) using a computer-based automatic device, until eight weeks after parturition. Colostrum yield was recorded within 3 h after parturition for three consecutive days.

#### 2.3.4. Milk and Colostrum Composition Analysis

During the experiment period, milk samples (03:00 and 15:00) were collected every two weeks from each individual for composition analysis. Following parturition (within 3 h), the colostrum was sampled in a 50 mL tube and analyzed immediately. The samples were analyzed for protein, fat, lactose, solid-not-fat (SNF), somatic cells, milk urea nitrogen (MUN), acetone, beta-hydroxybutyrate (BHB), beta-casein, mono- and polyunsaturated fatty acids, and saturated fatty acids using a milk scanner FT1 (Foss Alle 1 DK-3400 Hilleroed, Denmark). The milk fat yield, milk protein yield, 3.5% fat corrected milk (FCM), and energy corrected milk (FCM) were calculated by multiplying the milk yield by the protein and fat composition of the milk of an individual.

#### 2.3.5. Blood Analyses

Blood samples to obtain serum were collected every two weeks over the six weeks prepartum in dry cows and eight weeks postpartum in cows and calves via jugular venipuncture prior to the morning feeding at 08:30. Blood for serum (BD Vacutainer, Plymouth, UK) was collected, then centrifuged (20 °C, 15 min at 2000× *g*) and stored at −80 °C until further analyses. To analyze the complete blood count (CBC), the blood samples were poured into ethylene diamine-tetra-acetic acid (EDTA)-treated vacutainers (Becton-Dickinson, Franklin Lakes, NJ, USA) to collect whole blood for measuring white blood cells (WBC), lymphocytes, monocytes, granulocytes, the mean red blood cell numbers (RBC), hemoglobin, hematocrit, corpuscular volume (MCV), RDWc, mean corpuscular hemoglobin (MCH), mean corpuscular hemoglobin concentration (MCHC), platelets, mean platelet volume (MPV), plateletcrit (PCT), platelet distribution width (PDWc), percent granulocytes, percent lymphocytes, and percent monocytes. The complete blood counts were immediately determined after blood collection using a hematology analyzer (VetScan HM2 Hematology System, Union City, CA, USA). Serum biochemical parameters, including albumin, glutamic-oxaloacetic transaminase (GOT), glutamic pyruvic transaminase (GPT), blood urea nitrogen (BUN), creatine (CREA), TG, cholesterol (CHO), glucose (GLC), high-density lipoproteins (HDL), low-density lipoproteins (LDL), calcium (CA), IP, magnesium (MG), and non-esterified fatty acid (NEFA) were analyzed using a chemical analyzer (HITACHI Automatic Analyzer Model 7180, Hitachi, Gyeonggi-do, South Korea).

### 2.4. Statistical Analysis

The body weight of the calves and colostrum data were subjected to student’s *t*-test comparisons (two-tailed) between the means of the different groups. Feed intake, milk composition, and blood analysis data were conducted based on repeated measure analysis using the mixed procedure of SAS (Studio Version, SAS Institute Inc., Cary, NC, USA) [13]. For each variable, four covariance structures (compound symmetry, autoregressive order 1, unstructured covariance, and variance components) were evaluated. The covariance structure that resulted in the lowest value for the Akaike information criterion [10] was selected to compare the corrected mean values for each trait. The data were used for repeated measurement analysis to investigate the interaction effect (diet × week). The distribution of the animals by treatment was considered as a random effect. The continuous variable (covariate) was removed from the model. Differences between the two subsets of data were considered statistically significant at *p*-values less than 0.05, and values between 0.05 and 0.10 were considered to have a significant trend tendency.

## 3. Results

### 3.1. Feed Intake

There was an increase (*p* < 0.05) in the FI of dry cows in the ACT-treated group over six weeks compared to the control group (Table 2). In addition, the interaction between diet and week showed a statistical trend (*p* = 0.0920).

### 3.2. Birth Weight and Weight Changes in Newborn Calves

The results showed that the birth weight was not different (*p* > 0.10) between the two groups. However, the newborn calves in the ACT-treated group were heavier than the calves in the control group, and the average difference in all BW measurements were maintained at about 5.5 kg continuously during the eight weeks after parturition (Table 3).

### 3.3. Milk Yield and Compositions

#### 3.3.1. Milk Yield

Milk production was not different (*p* > 0.10) between the two groups. However, an average higher yield of ~4 kg in the ACT group compared to the control group was maintained during the study.

#### 3.3.2. Milk Composition

During the 8-week postpartum period, the ACT-treated group showed no differences in milk composition (*p* > 0.10), including milk protein, milk fat, lactose, SNF, MUN, acetone, BHB, beta-casein, monounsaturated fatty acid, polyunsaturated fatty acid, saturated fatty acid, milk fat yield, milk protein yield, 3.5% FCM, and ECM compared to the control group (*p* > 0.10, Table 4). Saturated fatty acid in the colostrum was lower (*p* < 0.05) in the ACT-treated group than in the control group. Colostrum fat (%) tended to be lower (*p* = 0.0797) and protein yield tended to be higher (*p* = 0.0871) in the ACT-treated group than in the control diet group at the time of birth (Table 5). In addition, there were no differences (*p* > 0.10) in protein, fat, lactose, SNF, MUN, acetone, BHB, beta-casein, mono- and polyunsaturated fatty acids, saturated fatty acids, fat yield, protein yield, 3.5% FCM, and ECM between the two groups (Table 5).

### 3.4. Blood Characteristics

#### 3.4.1. Blood Characteristics of Dry Cows Pre- and Postpartum

The RDWc (*p* = 0.0633) and granulocytes (*p* = 0.0823) tended to be lower in the ACT-treated group than in the control group. Supplementation with ACT did not affect WBCs, lymphocytes, monocytes, granulocytes, RBCs, hemoglobin, hematocrit, MCV, MCH, MCHC, platelets, MPV, PCT, PDWc, percent lymphocytes, and percent monocytes in the blood (*p* > 0.10, Table 6). ACT did not affect any of the blood characteristics listed above compared to the control group (*p* > 0.10, Table 7). Analysis of blood metabolite changes during the 6-week prepartum period showed increased GOT and decreased TG in the ACT-treated group compared to the control group (*p* < 0.05). In addition, MG showed a tendency to be higher (*p* = 0.0934) in the ACT group than in the control group. In contrast, ACT supplementation did not affect albumin, GPT, BUN, CREA, CHO, GLC, HDL, LDL, CA, IP, or NEFA in the blood (*p* > 0.10, Table 8). Postpartum cows tended to have higher LDL (*p* = 0.0551) and lower (*p* = 0.0966) IP in the ACT-treated group compared to the control group. However, albumin, GOT, GPT, BUN, CREA, TG, CHO, GLC, HDL, CA, MG, and NEFA were not affected by ACT supplementation (*p* > 0.10, Table 9).

#### 3.4.2. Blood Characteristics in Calves

Analysis of the blood characteristics in calves showed a significant difference in monocyte levels, which were lower in the ACT treatment group compared to the control group (*p* < 0.05). MCHC tended to be lower in the ACT-treated group compared to the control group (*p* = 0.0547). However, the WBCs, lymphocytes, granulocytes, RBCs, hemoglobin, hematocrit, MCV, RDWc, MCH, platelets, MPV, PCT, PDWc, percent granulocytes, and percent lymphocytes were not affected by ACT supplementation to the corresponding group of dry cows (*p* > 0.10, Table 10). GLC was lower (*p* < 0.05) in the calves whose dams received ACT (Table 11). Albumin, GOT, GPT, BUN, CREA, TG, CHO, HDL, LDL, CA, IP, MG, and NEFA were not different between the two groups (*p* > 0.10, Table 11).

## 4. Discussion

Improving the energy balance postpartum can lower the incidence of disease, decrease the risk of lowered milk production [14], and maintain general health. Therefore, increasing the FI during the dry period is very important for the productivity of postpartum dairy cows [15], which was supported by feeding ACT in this study. A higher FI of ACT-supplemented feed was observed compared to the control group. One reason for the increased FI in the ACT-treated group could be attributed to the high rate of nutrient absorption in the small intestine. In a previous study in our laboratory [4], feeding 191.1 mg/kg of ACT to beef cattle decreased the fecal flow of starch intake and increased the disappearance (57.8 mg/kg BW, 72%) of starch in the total tract. In addition, Vazquez-Anon et al. [16] demonstrated that the accumulation of TG caused a negative energy balance and reduced feed intake in dairy cows. Consistent with the aforementioned study, the reduced FI of the control group in the present study was also triggered by higher serum TG levels in prepartum cows (Table 8). Given the above discussion, higher FI due to ACT treatment positively affected FI, with an average of 0.46 kg in the treatment group (7.36 kg) compared to the control (6.90 kg). Additionally, and supporting our results, ACT was shown to be an optimized product that combined acetate and L-tryptophan to minimize digestion by microorganisms in the rumen, by bypassing it, facilitating digestion and absorption in the small intestine [5]. Therefore, this suggests that the supply of rumen-protected tryptophan may have increased FI because ACT promoted intestinal digestion and intracellular metabolism.

The cumulative effects of the treatment after eight weeks and at weaning were observed in the BW of calves whose dams received ACT treatment. L-tryptophan metabolites affect the growth, development, reproduction, and health of animals [17]. Generally, AAs are insufficient for performing specific physiological functions, and feeding rumen-protected AA forms was shown to have beneficial effects on the growth of lambs [12]. The higher birth weight of calves in the ACT-treated group of the respective dry cows could be attributed to higher FI due to ACT supplementation. Unbalanced nutrition during the gestation period of dry cows could reduce the birth weight of the calves and limit growth in the womb, through the carryover effect of absorption and utilization of nutrients [18]. This can be considered a positive effect of ACT in dry cows and compensatory effects on their offspring. Accordingly, we observed an approximately 5.5 kg higher BW of offspring from dams who received ACT treatment throughout the 8-week observation period. Further larger scale investigation on growing calves for more than eight weeks is warranted to evaluate the compensatory effects of ACT supplementation in dry cows through the performance of their offspring.

At the end of dry period and after parturition, the intake of L-tryptophan decreases due to lower feed intake. Moreover, L-tryptophan is an essential amino acid that is not sufficiently synthesized through microorganisms to achieve high production performance and animal growth. Thus, L-tryptophan should be supplemented to the diet to increase absorption from the diet and via microbial synthesis to meet the daily requirements for high production performance [6,7]. L-tryptophan together with other EAAs was reported to affect productivity, such as growth and milk production, in dairy cows [19]. We expected an increase in milk protein in cows postpartum from ACT supplementation. However, we could not find any difference in milk yield, nor in its composition, between the two groups. Milk protein synthesis is carried out through the mTOR pathway by L-tryptophan [20], but the percentage of milk protein in this study was not different between the two groups. In addition, lactose, SNF, lactose, acetone, monounsaturated FA, polyunsaturated FA, SFA, and MUN were not significantly affected by ACT supplementation (Table 4). The reason for the non-significant differences in milk yield and milk composition between the two groups could be attributed to the amount of ACT supplementation, which was limited to 15 g/day for each individual. Accordingly, Kollmann [21] stated that 125 g of rumen-protected tryptophan should be fed to cows to improve milk yield and composition. Another reason for not finding differences in milk yield and composition could be the lower sample size in this study. A further study with a larger sample size may confirm the hypothesis of this study, that ACT would increase milk yield and quality. Given the above reviews and our study, milk production was also quantitatively increased by ACT supplementation of dry cows for eight weeks postpartum, where the difference in FI was significantly higher in the ACT-treated group during the dry period.

In the offspring, since IgG is not transmitted to the calves through the placenta, the postpartum component in the colostrum is pivotal for calves’ health [22]. Colostrum containing IgG must be fed to calves within 24 h postpartum to improve immunity, which will have a positive impact on calves’ growth, health, and eventually, productivity [23]. Thus, one aim of feeding ACT to dry cows was to potentially increase colostrum protein and immunity, to positively affect the health of offspring. In this study, the percentage of milk fat in the colostrum tended to decrease and the milk protein yield tended to increase in the ACT-treated group (Table 5). Energy is needed to produce proteins and fats in milk. Thus, more energy was used to produce milk protein than milk fat in the ACT-treated group. As the amount of colostrum increased, the milk protein increased. Therefore, the increase in total milk protein and yield through the increase of colostrum yield may be attributed to the supply of more ACT by higher FI during the dry period, which could also influence the development of the mammary glands and eventually increase the amount of colostrum. The decomposed acetate produces energy through milk fat synthesis and the TCA cycle to provide the energy needed for milk production, and increased milk fat increases the FCM [24]. L-tryptophan addition to incubation medium for phospholipid biosynthesis in vitro caused decreases in the SFA content of phospholipids. Therefore, in our study, we assumed that the form of rumen-protected AA may have reduced the FAs in milk. The authors suggested that milk protein synthesis in dairy cows relied on the transfer of a sufficient supply of each essential AA, such as L-tryptophan, to the mammary gland. The lack of change in other milk components, including lactose, solid-not-fat, acetone, and milk urea nitrogen, implies that ACT did not improve those parameters in milk. This non-significant influence of ACT might have been due to the low supplementation (only 15 g/day) or sample size. Further studies with higher amounts of ACT supplementation and larger sample sizes might increase the differences.

Generally, the degree of occurrence of health problems during the dry period is attributed to changes in the DMI, and the response to treatment changes [25] can be observed through blood parameters. Nutritional deficiencies of L-tryptophan led to a decrease in the immune response [26]. This phenomenon may stimulate the immune response, leading to an increase in RDWc, as observed in the present study (Table 6). RDWc has been used to diagnose anemia for a long time [27]. Consistent with our study, which showed a tendency toward increased RDWc in dry cows prepartum, increased RDWc has been used as an indicator of anisocytosis, nutrient deficiency, and a large number of disorders [27] (Table 6). No statistical differences in the RDWc of offspring were seen, probably because ACT supplementation (only 15 g/day) to the respective dams or the sample size (*n* = 4) was not sufficient to see a carry-over effect of ACT. The FI also increased when CHO levels were increased in animals [28]. However, in this study, we did not find any significant differences in CHO levels in the ACT-treated group compared to the control group (Table 8). With respect to MG, when MG-deficient diets were fed to ruminants, FI was reduced [29]. In agreement with this event, in the present study, the ACT-treated group had a tendency toward higher MG levels than the control group (Table 8). It has been reported that the relatively low GLC and high NEFA levels in dry period cows were related to the increased incidence of fetal growth membrane retention [30]. However, no differences were observed in serum GLC, albumin, CHO, and NEFA, either prepartum or postpartum, in either group. This may be because the quantity of ACT added was insufficient (15 g/day/cow). Subsequent studies with higher amounts of ACT supplementation might yield different results. During the prepartum experiment (Table 8), there was a significant difference in the GOT content, which was higher in the ACT-treated group. The metabolic function of GOT is essentially related to the main energy supply pathway of mitochondrial enzyme activity [31]. Increased GOT activity in a porcine muscle also increased the muscle pigment content. Induction of the rapamycin (mTOR) pathway is important for the role of AAs in the synthesis of muscle proteins in the skeletal muscles of humans and rodents. The mTOR pathway is a key factor regulating muscle protein synthesis and AAs [32]. In support of this result, L-tryptophan is suggested to be transformed into acetyl-CoA and pyruvate in the TCA cycle, which occurs in the liver [33]. As discussed earlier, higher FI could have resulted in lower TG levels, as observed in the ACT treatment group. TG is an ester bond of glycerol and fatty acid. It also plays an important role as an energy source for ruminants. TG is stored in the liver and subcutaneous tissues. When sugar is lacking as an energy source, TG is hydrolyzed to mono glycerol, glycerol, and free fatty acid from fat decomposition, and released into the blood and used as an energy source [34]. TG testing in animals is mainly used to identify problems with lipid metabolism. Dietary tryptophan reduces the concentration of TG. The reduction in TG cannot be attributed to an impaired lipoprotein output from the liver, although other research noted an effect at a much higher dose of tryptophan [35]. The results of this study also showed a statistical difference in TG in the prepartum cows (Table 8). The results indicated lower lipid metabolism in the ACT-treated group compared to the control group. The most important role of LDL is to deliver CHO to the peripheral tissues. A previous study showed that LDL levels increased by feeding rabbits an essential AA compound mixture [36]. Our experiment similarly showed a tendency toward higher LDL levels in postpartum cows that received ACT supplementation (Table 9). The early lactation period is associated with changes in the IP metabolic process [37]. In pigs, the blood phosphorus levels decreased by feeding a diet containing 18% protein compared to 12% protein [38]. Consistent with those findings, the results of this study suggest that the addition of ACT may have increased dietary protein, resulting in a decrease in postpartum serum phosphorus (Table 9).

The in vivo ACT treatment experiment showed changes in the blood characteristics of calves, which included differences in monocytes (Table 10) and effects on the MCHC. Monocytes are part of the mononuclear phagocytic system that affects the growth, inflammation, and immune systems. This indicates that ACT supplementation had a positive effect on calf growth because it regulated the inflammatory response mechanism and suppressed the growth of pathogens [39]. The relative MCHC amounts tended to be higher in the ACT-treated group than in the control group. Less hemoglobin means less oxygen transportation. Oxygen transport has a positive effect on nutrition and metabolism [40]. Our results suggest that the immune response affected hemoglobin levels in the erythrocytes of calves whose dams were fed ACT during the experimental period. Given the blood biochemical results, cellulose is digested in the rumen to products similar to those of starch digestion, including a volatile fatty acid called propionate, which eventually can be partially used for GLC production in the liver. GLC is transported to each tissue and used as an energy source or stored as glycogen and used when necessary [26]. Tryptophan contributes to acetyl-CoA and acetoacetyl-CoA, which affect the citric acid cycle, resulting in energy metabolism control. The serum GLC levels in the ACT-treated group of calves (Table 11) were lower than those in the control group. The lower levels of serum GLC in ACT-treated calves may have been due to the increased starch digestibility. This means that GLC was absorbed in the small intestine and affected the growth performance of the calves. In contrast, tryptophan supplementation had no significant effects on blood GLC levels in poultry [41]. It was thought that the unchanged blood biochemical components in the calves could have been due to the low amount of ACT supplementation to their dams and the subsequent low transfer through the placenta. Therefore, further study is needed to investigate the effect of feeding L-tryptophan to dry cows and to observe its effect on newborn calves fed supplemental AA compared to non-supplemented calves.

## 5. Conclusions

In conclusion, the experimental results showed that acetate and L-tryptophan-conjugated bypass amino acid (ACT), positively affected the FI of prepartum cows, the weight of newborn calves, and milk protein yield in postpartum cows. The results implied a positive nutrient compensation by ACT treatment for decreased appetite and further production disturbances that occurred from hormonal imbalances in the dry period of dairy cows. Overall, ACT as a feed additive has positive contributions in dry period cows, favors subsequent lactation, and has positive carry-over effects on the birth weight of offspring. Further large-scale studies with various ACT supplementation amounts are warranted to confirm this conclusion. We also suggest rearing newborn calves up to weaning to track the possible carry-over effects of ACT supplementation on their growth performance.

## Figures and Tables

**Table 1 animals-11-01726-t001:** Compositions of basal diet and ACT used in this study (DM basis).

Diets	Control	Treatment
TMR, kg/day	1.0	15.0
Roughage, kg/day	9.8	
Concentrates, kg/day	5.0	
ACT, g/day	0.0	
Chemical compositions, g/day on DM basis		
Dry Matter		
Crude Protein	1529.92	
Crude Fat	213.36	
Crude Fiber	2558.43	
Crude Ash	1008.48	
Calcium	92.25	
Phosphorus	39.48	
Amino acids, g/day on Dry Matter basis		
Tryptophan	7.39	22.39
Threonine	49.76	
Serine	55.29	
Proline	84.34	
Valine	70.60	
Isoleucine	46.07	
Leucine	95.49	
Tyrosine	29.20	
Methionine	27.89	
Cysteine	40.87	
Lysine	56.02	
Glycine	63.56	
Alanine	74.71	
Arginine	78.16	
Glutamic Acid	187.09	
Aspartic Acid	130.43	
Histidine	27.75	
Phenylalanine	57.03	

TMR, total mixed ration; ACT, acetate and L-tryptophan-conjugated bypass amino acid.

**Table 2 animals-11-01726-t002:** Average dry matter intake (DMI) of dry cows.

Item	Treatment ^1^		SEM	*p*-Value		
	Control	ACT		D	W	D × W
Feed intake, kg/day	6.90	7.36	0.144	0.0420	0.5760	0.0920

Values are expressed as means (*n* = 4). ^1^ Treatment = control, basal diet; ACT, 15 g per day of basal diet. D, diet; W, week; ACT, acetate and L-tryptophan-conjugated bypass amino acid; SEM, standard error mean.

**Table 3 animals-11-01726-t003:** Weight of calves birthed by dry cows fed basal diet (control) and diet containing ACT for 8 weeks.

Item	Treatment ^1^		SEM	*p*-Value
	Control	ACT		
Body weight (kg)				
0 week	35.90	41.43	3.063	0.4086
2 week	40.95	46.05	3.085	0.4512
4 week	48.08	52.75	3.381	0.5319
6 week	56.73	60.95	3.601	0.5977
8 week	63.80	69.70	3.609	0.4565

Values are expressed as means (*n* = 4). ^1^ Treatment = control, basal diet; ACT, 15 g per day of basal diet. ACT, acetate and L-tryptophan-conjugated bypass amino acid; SEM, standard error mean. Data are the average body weight.

**Table 4 animals-11-01726-t004:** Impact of ACT on average milk yield and composition during 8-week treatment of dry cows postpartum.

Item ^2^	Treatment ^1^		SEM	*p*-Value		
	Control	ACT		D	W	D × W
Milk yield, kg/day	31.77	35.44	2.401	0.9343	0.0020	0.0956
Milk protein, %	4.83	4.88	0.266	0.9343	0.0020	0.0956
Milk fat, %	6.62	6.31	0.380	0.7125	0.0004	0.0017
Lactose, %	4.37	4.38	0.036	0.9398	<0.0001	0.9635
Solid-not-fat, %	9.88	9.95	0.259	0.9056	<0.0001	0.9850
Milk urea nitrogen, mg/dL	16.29	16.49	0.578	0.8778	<0.0001	0.4981
Acetone, mM	0.04	0.07	0.025	0.4170	0.0695	0.8466
BHB, mM	0.03	0.04	0.010	0.4892	0.0001	0.7592
Cas.B, %	3.66	3.71	0.207	0.9134	0.0014	0.4718
Mono FA, %	1.74	1.70	0.162	0.9082	0.0013	0.0007
Poly FA, %	0.36	0.36	0.010	0.8059	0.0005	0.4966
Saturated FA, %	4.51	4.20	0.217	0.5668	<0.0001	0.0006
Milk fat yield, kg/d	2.11	2.28	0.252	0.7617	0.0072	0.0182
Milk protein yield, kg/d	1.27	1.50	0.099	0.2791	<0.0001	0.3215
3.5% FCM, kg/d	47.97	52.33	5.014	0.6655	0.0007	0.3117
ECM, kg/d	46.89	51.94	4.487	0.5815	<0.0001	0.5331

Values are expressed as means (*n* = 4). ^1^ Treatment = control, basal diet; ACT, 15 g per day of basal diet. Data are the average milk yield and composition for 8 weeks. ^2^ Abbreviations = ACT, acetate and L-tryptophan-conjugated bypass amino acid; BHB, beta-hydroxybutyrate; Cas B, beta-casein; Mono FA, mono-unsaturated fatty acid; Poly FA, poly-unsaturated fatty acid; FCM, fat corrected milk; ECM, energy corrected milk. D, diet; W, week; SEM, standard error mean.

**Table 5 animals-11-01726-t005:** Impact of ACT on colostrum yield and composition in cows postpartum.

Item ^2^	Treatment ^1^		SEM	*p*-Value
	Control	ACT		
Colostrum yield, kg	19.83	24.03	1.894	0.3012
Protein, %	12.20	12.81	1.219	0.8259
Fat, %	9.64	6.25	0.985	0.0797
Lactose, %	3.36	3.48	0.108	0.6340
Solid-not-fat, %	16.35	17.17	1.153	0.7531
Urea nitrogen, mg/dL	23.80	24.43	1.735	0.8727
Acetone, mM	0.00	0.00	0.000	-
BHB, mM	0.00	0.00	0.000	-
Cas.B, %	9.32	9.76	0.943	0.8363
Mono FA, %	2.50	1.35	0.505	0.2868
Poly FA, %	0.39	0.34	0.042	0.5962
Saturated FA, %	7.18	4.95	0.541	0.0228
Fat yield, kg	2.01	1.48	0.282	0.3775
Protein yield, kg	2.23	3.04	0.237	0.0871
3.5% FCM, kg	41.24	34.33	5.042	0.5354
ECM, kg	48.65	48.85	4.150	0.9829

Values are expressed as means (*n* = 4). ^1^ Treatment = control, basal diet; ACT, 15 g per day of basal diet. ^2^ Abbreviations = ACT, acetate and L-tryptophan-conjugated bypass amino acid; BHB, beta-hydroxybutyrate; Cas B, beta-casein; Mono FA, mono-unsaturated fatty acid; Poly FA, poly-unsaturated fatty acid; FCM, fat corrected milk; ECM, energy corrected milk; SEM, standard error mean. Colostrum yield was measured daily by milking the animals twice a day (morning and afternoon) for 3 consecutive days prior to the colostrum sampling and feeding to the calves. The average colostrum yield was measured. The time of measurement, sampling, and feeding of colostrum was the same for the groups, to avoid confounded results.

**Table 6 animals-11-01726-t006:** Complete blood count analysis of cows prepartum.

Item ^2^	Treatment ^1^		SEM	*p*-Value		
	Control	ACT		D	W	D × W
WBCs, 10^9^/L	9.04	11.91	1.530	0.4509	0.0248	0.0253
Lymphocytes, 10^9^/L	4.52	7.88	1.453	0.3104	0.0487	0.2047
Monocytes, 10^9^/L	0.69	0.74	0.050	0.5102	0.0112	0.1461
Granulocytes, 10^9^/L	3.83	3.29	0.215	0.2689	0.0194	<0.0001
RBCs, 10^12^/L	7.16	7.18	0.155	0.9016	0.0322	0.3924
Hemoglobin, g/dL	11.61	12.03	0.355	0.4813	0.0397	0.0651
Hematocrit, %	33.55	34.21	0.840	0.6097	0.0303	0.3662
MCV, fL	46.85	47.54	0.794	0.7488	<0.0001	<0.0001
RDWc, %	19.86	18.77	0.296	0.0633	0.8178	0.6203
MCH, pg	16.20	16.74	0.269	0.2808	0.0012	0.0181
MCHC, g/dL	34.59	35.13	0.284	0.3260	0.0070	0.0480
Platelet, 10^9^/L	294.73	331.35	20.419	0.2999	0.0335	0.2348
MPV, fL	8.11	8.18	0.164	0.7732	0.0006	0.9459
PCT, %	0.24	0.27	0.020	0.3997	0.0095	0.2052
PDWc, %	35.15	35.16	0.340	0.9865	<0.0001	0.5421
GR, %	41.70	30.53	3.225	0.0823	0.5225	<0.0001
LY, %	50.98	62.40	3.594	0.1217	0.2481	0.0011
MO, %	7.32	7.09	0.784	0.8403	0.0481	0.7762

Values are expressed as means (*n* = 4). ^1^ Treatment = control, basal diet; ACT, 15 g per day of basal diet. ^2^ Abbreviations = ACT, acetate and L-tryptophan-conjugated bypass amino acid; WBC, white blood cell; RBC, red blood cell; MCV, mean corpuscular volume; RDWc, red cell distribution width; MCH, mean corpuscular hemoglobin; MCHC, mean corpuscular hemoglobin concentration; MPV, mean platelet volume; PCT, plateletcrit; PDWc, platelet distribution width; GR, granulocyte percent; LY, lymphocyte percent; MO, monocyte percent; D, diet; W, week; SEM, standard error mean.

**Table 7 animals-11-01726-t007:** Complete blood count analysis of cows postpartum.

Item ^2^	Treatment ^1^		SEM	*p*-Value		
	Control	ACT		D	W	D × W
WBCs, 10^9^/L	10.10	13.95	1.554	0.2422	0.0393	0.7195
Lymphocytes, 10^9^/L	4.98	7.86	1.438	0.3559	0.6989	0.1867
Monocytes, 10^9^/L	0.85	0.91	0.057	0.6888	0.6692	0.7586
Granulocytes, 10^9^/L	4.27	5.18	0.414	0.3032	0.0851	0.5479
RBCs, 10^12^/L	6.85	6.67	0.181	0.6518	<0.0001	0.0188
Hemoglobin, g/dL	10.95	11.03	0.220	0.8714	<0.0001	0.0840
Hematocrit, %	30.89	30.96	0.567	0.9554	<0.0001	0.3311
MCV, fL	45.20	46.55	0.869	0.4206	0.0178	0.0047
RDWc, %	19.63	19.28	0.160	0.2999	<0.0001	<0.0001
MCH, pg	16.03	16.60	0.241	0.2659	0.0044	0.0773
MCHC, g/dL	35.50	35.68	0.231	0.7276	0.0003	0.0324
Platelet, 10^9^/L	447.15	480.51	26.278	0.5672	<0.0001	0.0023
MPV, fL	7.31	7.47	0.158	0.6576	0.2457	0.2282
PCT, %	0.33	0.37	0.024	0.4012	<0.0001	0.5041
PDWc, %	32.76	32.77	0.368	0.9856	0.0039	0.0104
GR, %	41.07	40.77	3.904	0.9728	0.6534	0.6911
LY, %	50.66	52.00	4.198	0.8876	0.3492	0.0353
MO, %	8.27	7.24	0.817	0.5679	0.6838	0.5729

Values are expressed as means (*n* = 4). ^1^ Treatment = control, basal diet; ACT, 15 g per day of basal diet. ^2^ Abbreviations = ACT, acetate and L-tryptophan-conjugated bypass amino acid; WBC, white blood cell; RBC, red blood cell; MCV, mean corpuscular volume; RDWc, red cell distribution width; MCH, mean corpuscular hemoglobin; MCHC, mean corpuscular hemoglobin concentration; MPV, mean platelet volume; PCT, plateletcrit; PDWc, platelet distribution width; GR, granulocyte percent; LY, lymphocyte percent; MO, monocyte percent; D, diet; W, week; SEM, standard error mean.

**Table 8 animals-11-01726-t008:** Serum metabolic profile of cows prepartum.

Item ^2^	Treatment ^1^		SEM	*p*-Value		
	Control	ACT		D	W	D × W
Albumin (g/dL)	3.68	3.74	0.035	0.5019	0.0058	0.1369
GOT (U/L^3^)	68.56	80.92	3.459	0.0055	0.4605	0.5441
GPT (U/L^3^)	21.19	20.88	1.070	0.7741	0.0087	0.8094
BUN (mg/dL)	11.10	9.88	0.616	0.3729	0.0022	0.6781
CREA (mg/dL)	1.59	1.58	0.064	0.9675	0.0294	0.0503
TG (mg/dL)	22.46	19.88	1.134	0.0084	0.0126	0.0198
CHO (mg/dL)	129.90	143.08	6.949	0.5735	0.0015	0.3177
GLC (mg/dL)	54.04	50.52	1.837	0.4163	<0.0001	0.3098
HDL-cholesterol (mg/dL)	76.42	80.88	4.141	0.9428	<0.0001	0.0312
LDL-cholesterol (mg/dL)	18.21	21.38	2.654	0.6751	0.0003	0.5950
CA (mg/dL)	9.47	9.21	0.266	0.6581	0.0003	0.1866
IP (mg/dL)	5.58	6.41	0.251	0.1437	0.0272	0.0461
MG (mg/dL)	2.44	2.64	0.060	0.0934	<0.0001	0.0186
NEFA (mmol/L)	399.60	480.13	70.863	0.6545	0.5193	0.0410

Values are expressed as means (*n* = 4). ^1^ Treatment = control, basal diet; ACT, 15 g per day of basal diet. ^2^ Abbreviations = ACT, acetate and L-tryptophan-conjugated bypass amino acid; GOT, glutamic-oxaloacetic transaminase; GPT, glutamic pyruvic transaminase; BUN, blood urea nitrogen; CREA, creatine; TG, triglycerides; CHO, cholesterol; GLC, glucose; HDL, high-density lipoproteins; LDL, low-density lipoproteins; CA, calcium; IP, inorganic phosphorus; MG, magnesium; NEFA, non-esterified fatty acid; D, diet; W, week; SEM, standard error mean. ^3^ 1U (µmol/min) is defined as the amount of enzyme activity that catalyzes the conversion of one micromole of substrate per minute under the specified conditions of the assay method.

**Table 9 animals-11-01726-t009:** Serum metabolic profile of cows postpartum.

Item ^2^	Treatment ^1^		SEM	*p*-Value		
	Control	ACT		D	W	D × W
Albumin (g/dL)	3.88	4.01	0.097	0.5470	<0.0001	0.7086
GOT (U/L ^3^)	83.55	89.55	2.388	0.2345	0.0578	0.1285
GPT (U/L ^3^)	23.70	22.40	1.389	0.6764	<0.0001	0.6904
BUN (mg/dL)	17.95	16.75	0.784	0.4788	0.0007	0.0422
CREA (mg/dL)	1.02	1.01	0.028	0.8480	<0.0001	0.8612
TG (mg/dL)	9.05	9.45	0.401	0.6379	0.0091	0.9999
CHO (mg/dL)	164.10	188.45	10.093	0.2440	<0.0001	0.5058
GLC (mg/dL)	75.20	68.35	3.104	0.3040	0.0592	0.5389
HDL-cholesterol (mg/dL)	115.25	125.60	4.262	0.2140	<0.0001	0.7894
LDL-cholesterol (mg/dL)	14.65	19.35	1.364	0.0551	<0.0001	0.3174
CA (mg/dL)	9.63	9.20	0.226	0.3903	0.0218	0.0497
IP (mg/dL)	5.95	5.30	0.351	0.0966	0.3489	0.2723
MG (mg/dL)	2.94	3.15	0.082	0.2225	<0.0001	0.0469
NEFA (mmol/L)	198.49	301.27	38.480	0.2020	0.0079	0.0291

Values are expressed as means (*n* = 4). ^1^ Treatment = control, basal diet; ACT, 15 g per day of basal diet. ^2^ Abbreviations = ACT, acetate and L-tryptophan-conjugated bypass amino acid; GOT, glutamic-oxaloacetic transaminase; GPT, glutamic pyruvic transaminase; BUN, blood urea nitrogen; CREA, creatine; TG, triglycerides; CHO, cholesterol; GLC, glucose; HDL, high-density lipoproteins; LDL, low-density lipoproteins; CA, calcium; IP, inorganic phosphorus; MG, magnesium; NEFA, non-esterified fatty acid; D, diet; W, week; SEM, standard error mean. ^3^ 1U (µmol/min) is defined as the amount of enzyme activity that catalyzes the conversion of one micromole of substrate per minute under the specified conditions of the assay method.

**Table 10 animals-11-01726-t010:** Complete blood count analysis in calves.

Item ^2^	Treatment ^1^		SEM	*p*-Value		
	Control	ACT		D	W	D × W
WBCs, 10^9^/L	11.02	12.14	0.320	0.1810	0.9608	0.2260
Lymphocytes, 10^9^/L	6.43	6.93	0.448	0.6390	<0.0001	0.0192
Monocytes, 10^9^/L	0.82	0.53	0.074	0.0198	<0.0001	<0.0001
Granulocytes, 10^9^/L	3.77	4.69	0.418	0.3059	0.0004	0.0460
RBCs, 10^12^/L	10.29	10.13	0.281	0.7976	0.0029	0.8783
Hemoglobin, g/dL	11.22	11.28	0.541	0.9607	<0.0001	0.1210
Hematocrit, %	31.35	32.16	1.503	0.7959	<0.0001	0.9200
MCV, fL	30.45	31.80	0.683	0.9392	<0.0001	0.6488
RDWc, %	26.19	25.08	0.781	0.5167	0.0164	0.6422
MCH, pg	10.89	11.20	0.261	0.3378	<0.0001	0.1710
MCHC, g/dL	35.86	35.18	0.183	0.0547	<0.0005	0.2394
Platelet, 10^9^/L	515.40	584.35	53.237	0.5591	0.0310	0.0889
MPV, fL	6.21	6.57	0.222	0.4604	0.0089	0.0549
PCT, %	0.32	0.37	0.033	0.4551	0.0935	0.0197
PDWc, %	30.49	30.36	0.684	0.9328	0.0088	0.0619
GR, %	34.01	37.06	3.243	0.6529	0.0002	0.0225
LY, %	58.73	58.30	3.527	0.9553	0.0008	0.0363
MO, %	7.26	4.62	0.684	0.0277	<0.0001	<0.0001

Values are expressed as means (*n* = 4). ^1^ Treatment = control, basal diet; ACT, 15 g per day of basal diet. ^2^ Abbreviations = ACT, acetate and L-tryptophan-conjugated bypass amino acid; WBC, white blood cell; RBC, red blood cell; MCV, mean corpuscular volume; RDWc, red cell distribution width; MCH, mean corpuscular hemoglobin; MCHC, mean corpuscular hemoglobin concentration; MPV, mean platelet volume; PCT, plateletcrit; PDWc, platelet distribution width; GR, granulocyte percent; LY, lymphocyte percent; MO, monocyte percent; D, diet; W, week; SEM, standard error mean.

**Table 11 animals-11-01726-t011:** Serum metabolic profile in calves.

Item ^2^	Treatment ^1^		SEM	*p*-Value		
	Control	ACT		D	W	D × W
Albumin (g/dL)	3.44	3.36	0.091	0.6958	<0.0001	0.4439
GOT (U/L ^3^)	65.00	62.55	2.623	0.6722	<0.0001	0.3450
GPT (U/L ^3^)	11.95	12.25	0.676	0.8478	<0.0001	0.4106
BUN (mg/dL)	11.50	11.30	0.688	0.8950	0.0164	0.2005
CREA (mg/dL)	1.17	1.57	0.192	0.3305	0.0170	0.0549
TG (mg/dL)	37.95	34.40	4.570	0.7406	0.0040	0.2307
CHO (mg/dL)	98.35	109.30	6.935	0.4728	0.0002	0.6272
GLC (mg/dL)	97.85	78.95	4.416	0.0111	0.0152	0.6324
HDL-cholesterol (mg/dL)	76.80	77.55	4.706	0.9436	<0.0001	0.1293
LDL-cholesterol (mg/dL)	13.80	15.17	1.467	0.7113	0.0031	0.6633
CA (mg/dL)	10.31	10.31	0.107	1.0000	0.0007	0.0079
IP (mg/dL)	8.64	8.09	0.291	0.3606	<0.0001	0.4574
MG (mg/dL)	2.63	2.60	0.071	0.8515	0.0061	0.3852
NEFA (mmol/L)	219.64	210.55	17.321	0.8154	0.0040	0.8764

Values are expressed as means (*n* = 4). ^1^ Treatment = control, basal diet; ACT, 15 g per day of basal diet. ^2^ Abbreviations = ACT, acetate and L-tryptophan-conjugated bypass amino acid; GOT, glutamic-oxaloacetic transaminase; GPT, glutamic pyruvic transaminase; BUN, blood urea nitrogen; CREA, creatine; TG, triglycerides; CHO, cholesterol; GLC, glucose; HDL, high-density lipoproteins; LDL, low-density lipoproteins; CA, calcium; IP, inorganic phosphorus; MG, magnesium; NEFA, non-esterified fatty acid; D, diet; W, week; SEM, standard error mean. ^3^ 1U (µmol/min) is defined as the amount of enzyme activity that catalyzes the conversion of one micromole of substrate per minute under the specified conditions of the assay method.

## Data Availability

Not applicable.

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
