# Peer review of "Effects of Dietary Supplementation of Acetate and L-Tryptophan Conjugated Bypass Amino Acid on Productivity of Pre- and Post-Partum Dairy Cows and Their Offspring"

_animals, 2021, doi:10.3390/ani11061726_

Round 1

Reviewer 1 Report

Author’s made some additions, corrections and changes based ob reviewer’s suggestions, but still the basic drawbacks remain. There are substantial problems in the study design (sample size, treatment, sampling procedure and measurements) and presentation that make results not justified by methods and arise doubts about findings and conclusions.

Author Response

Open Review

English language and style

( ) Extensive editing of English language and style required
( ) Moderate English changes required
(x) English language and style are fine/minor spell check required
( ) I don't feel qualified to judge about the English language and style

Yes

Can be improved

Must be improved

Not applicable

Does the introduction provide sufficient background and include all relevant references?

(x)

( )

( )

( )

Is the research design appropriate?

( )

( )

(x)

( )

Are the methods adequately described?

( )

( )

(x)

( )

Are the results clearly presented?

( )

( )

(x)

( )

Are the conclusions supported by the results?

( )

( )

(x)

( )

Comments and Suggestions for Authors

Author’s made some additions, corrections and changes based ob reviewer’s suggestions, but still the basic drawbacks remain. There are substantial problems in the study design (sample size, treatment, sampling procedure and measurements) and presentation that make results not justified by methods and arise doubts about findings and conclusions.

Response: Thank you for your insights and comments. As the reviewer may know, studying on dry cows and tracking the carry-over effects on the offspring is a hot topic receiving a lot of attention from avid readers of high reputative journals. Thus, we hope that more considerations from the reviewers due to the importance, novelty, originality, and necessity of the study.

With regards to the concerns that the reviewer raised; We selected eight cows with the most similar conditions (parity, calving date, body weight) to conduct the experiment among the thirty cows in the farm. With being highly selective for choosing the experimental units (cows) from the pool of around 30 cows, we tried to decrease the errors rooted in experimental units. Plus, we also mentioned at different points in the Discussion regarding the limitation of this study for the number of experimental units used.

It should be noted that observing differences in performance criteria even with small size shows that the treatments were chosen carefully, and their effect were influential enough to overcome the individual difference between animals by showing the differences in results. One reason for choosing larger sample size is to reduce errors coming from individual differences. It is while we already tried to reduce the individual differences between cows of within and between groups by homogenizing the groups and less deviations in BW and parity of each group to cover this possible issue. Thus, we could statistically reduce the error among the individuals by strictly homogenizing the above-mentioned criteria.  As we wrote in the manuscript, the feed intake was measured daily and blood and milk sampling were collected once every two weeks which give us enough replications to draw the conclusion.  It worth noting that we already published papers with the exact same sample size and similar sampling procedures in other esteemed journals that we now list a few below for your kind consideration:

  1. Animals: Lee et al., (27 November 2019). “Effect of Dietary Rumen-Protected L-Tryptophan Supplementation on Growth Performance, Blood Hematological and Biochemical Profiles, and Gene Expression in Korean Native Steers under Cold Environment”. Animals. 9. 1036. doi:10.3390/ani9121036
  2. Journal of Thermal Biology: Dietary supplementation of acetate-conjugated tryptophan alters feed intake, milk yield and composition, blood profile, physiological variables, and heat shock protein gene expression in heat-stressed dairy cows (https://doi.org/10.1016/j.jtherbio.2021.102949).
  3. Journal of Animals Science and Technology: Wahyu et al., (20 June 2020). “Dietary supplementation of L-tryptophan increases muscle development, adipose tissue catabolism and fatty acid transportation in the muscles of Hanwoo steers”. Journal of Animals Science and Technology. 62(5). 595-604. https://doi.org/10.5187/jast.2020.62.5.595
  4. Asian-Australasian Journal of Animal Sciences (Animal Bioscience): Lee et al., (30 October 2019). “Administration of encapsulated L-tryptophan improves duodenal starch digestion and increases gastrointestinal hormones secretions in beef cattle”. Asian-Australasian Journal of Animal Sciences. 33. 91-99. https://doi.org/10.5713/ajas.19.0498

Once again, we are thankful to the reviewer for the insightful comments that all were taken into consideration to improve the quality of our presentation. Please find the revised version of our manuscript considering changes applied due to the reviewers’ and the Editor’s comments.

Reviewer 2 Report

Dear Authors, In fact, I found many changes in the revised version of the manuscript. The work is interesting and raises an important issue. She could bring interesting information to the area of high milk cow management. Unfortunately, it has too little scientific value because the experiment was carried out on a very small group of animals. The four cows in the experimental and control group are definitely a small population. Especially since the experiment is about introducing supplementation. Of course, the conditions that were met in the experimental plant eliminate many environmental factors. However, individual variability remains. There is no information on the origin of the cows and their homogeneity. The information obtained will be difficult to relate to a larger population,  especially in production conditions.

I propose to increase the experimental groups or to accurately describe the features proving the genetic homogeneity of the tested cows. This will strengthen the results and allow you to draw solid conclusions.

Author Response

Open Review

English language and style

( ) Extensive editing of English language and style required
( ) Moderate English changes required
(x) English language and style are fine/minor spell check required
( ) I don't feel qualified to judge about the English language and style

Yes

Can be improved

Must be improved

Not applicable

Does the introduction provide sufficient background and include all relevant references?

(x)

( )

( )

( )

Is the research design appropriate?

( )

( )

(x)

( )

Are the methods adequately described?

( )

(x)

( )

( )

Are the results clearly presented?

(x)

( )

( )

( )

Are the conclusions supported by the results?

( )

(x)

( )

( )

Comments and Suggestions for Authors

Dear Authors, In fact, I found many changes in the revised version of the manuscript. The work is interesting and raises an important issue. She could bring interesting information to the area of high milk cow management. Unfortunately, it has too little scientific value because the experiment was carried out on a very small group of animals. The four cows in the experimental and control group are definitely a small population. Especially since the experiment is about introducing supplementation. Of course, the conditions that were met in the experimental plant eliminate many environmental factors. However, individual variability remains. There is no information on the origin of the cows and their homogeneity. The information obtained will be difficult to relate to a larger population,  especially in production conditions.

I propose to increase the experimental groups or to accurately describe the features proving the genetic homogeneity of the tested cows. This will strengthen the results and allow you to draw solid conclusions.

Response: Thank you for the positive evaluation of our manuscript. As the reviewer mentioned in the comments, studying on dry cows and tracking the carry-over effects on the offspring is a hot topic receiving a lot of attention from vivid readers of high reputative journals. Thus, we hope that more considerations from the reviewers due to the importance, novelty, originality, and necessity of the study. We selected eight cows with the most similar conditions (parity, calving date, body weight) to conduct the experiment among the thirty cows in the farm. With being highly selective for choosing the experimental units (cows) from the pool of around 30 cows, we tried to decrease the errors rooted in experimental units. Plus, we also mentioned at different points in the Discussion regarding the limitation of this study for the number of experimental units used.

It should be noted that observing differences in performance criteria even with small size shows that the treatments were chosen carefully, and their effect were influential enough to overcome the individual difference between animals by showing the differences in results. One reason for choosing larger sample size is to reduce errors coming from individual differences. It is while we already tried to reduce the individual differences between cows of within and between groups by homogenizing the groups and less deviations in BW and parity of each group to cover this possible issue. Thus, we could statistically reduce the error among the individuals by strictly homogenizing the above-mentioned criteria.  As we wrote in the manuscript, the feed intake was measured daily and blood and milk sampling were collected once every two weeks which give us enough replications to draw the conclusion.  It worth noting that we already published papers with the exact same sample size and similar sampling procedures in other esteemed journals that we now list a few below for your kind consideration:

  1. Animals: Lee et al., (27 November 2019). “Effect of Dietary Rumen-Protected L-Tryptophan Supplementation on Growth Performance, Blood Hematological and Biochemical Profiles, and Gene Expression in Korean Native Steers under Cold Environment”. Animals. 9. 1036. doi:10.3390/ani9121036
  2. Journal of Thermal Biology: Dietary supplementation of acetate-conjugated tryptophan alters feed intake, milk yield and composition, blood profile, physiological variables, and heat shock protein gene expression in heat-stressed dairy cows (https://doi.org/10.1016/j.jtherbio.2021.102949).
  3. Journal of Animals Science and Technology: Wahyu et al., (20 June 2020). “Dietary supplementation of L-tryptophan increases muscle development, adipose tissue catabolism and fatty acid transportation in the muscles of Hanwoo steers”. Journal of Animals Science and Technology. 62(5). 595-604. https://doi.org/10.5187/jast.2020.62.5.595
  4. Asian-Australasian Journal of Animal Sciences (Animal Bioscience): Lee et al., (30 October 2019). “Administration of encapsulated L-tryptophan improves duodenal starch digestion and increases gastrointestinal hormones secretions in beef cattle”. Asian-Australasian Journal of Animal Sciences. 33. 91-99. https://doi.org/10.5713/ajas.19.0498

Once again, we are thankful to the reviewer for the insights comments that all were taken into consideration to improve the quality of our presentation. We tried to the best of our ability to provide logical response to the comments raised by the reviewer in here and reflected in the manuscript for the second revision. We particularly provided responses regarding the sample size, homogeneity of the cows with providing the related references. We truly hope that due to the novelty, and the hypotheses tested in the study, our revised manuscript can be satisfying the points raised by the reviewer.

Round 2

Reviewer 1 Report

Authors made substantial improvements and complied with reviewers’ recommendations. In the revised version material and methods are finally, sufficient and comprehensive. There are still language mistakes.

The basic drawback concerning the small number of animals and the few sampling points, especially concerning calves, remain and cannot change, but these points are clearly presented now.

Author Response

Open Review

English language and style

( ) Extensive editing of English language and style required
(x) Moderate English changes required
( ) English language and style are fine/minor spell check required
( ) I don't feel qualified to judge about the English language and style

Yes

Can be improved

Must be improved

Not applicable

Does the introduction provide sufficient background and include all relevant references?

(x)

( )

( )

( )

Is the research design appropriate?

( )

(x)

( )

( )

Are the methods adequately described?

(x)

( )

( )

( )

Are the results clearly presented?

(x)

( )

( )

( )

Are the conclusions supported by the results?

( )

(x)

( )

( )

Comments and Suggestions for Authors

Authors made substantial improvements and complied with reviewers’ recommendations. In the revised version material and methods are finally, sufficient and comprehensive. There are still language mistakes.

The basic drawback concerning the small number of animals and the few sampling points, especially concerning calves, remain and cannot change, but these points are clearly presented now.

Response: Thank you for considering our manuscript for publication in animals. We are thankful to the reviewer for the insights comments that all were taken into consideration to improve the quality of our presentation. We appreciate the time and effort that you and the reviewers have dedicated to providing your valuable feedback on our manuscript. Once again, we tried to the best of our ability to provide logical response to the comments raised by the reviewer in here and reflected in the manuscript revision. Particularly provided responses regarding the sample size, homogeneity of the cows with providing the related references. We truly hope that due to the novelty, and the hypotheses tested in the study, our revised manuscript can be satisfying the points raised by the reviewer.
